# Musculoskeletal-Modeling-Based, Full-Body Load-Assessment Tool for Ergonomists (MATE): Method Development and Proof of Concept Case Studies

**DOI:** 10.3390/ijerph20021507

**Published:** 2023-01-13

**Authors:** Arthur van der Have, Sam Van Rossom, Ilse Jonkers

**Affiliations:** Human Movement Biomechanics Research Group, Department of Movement Sciences, Katholieke Universiteit Leuven, 3001 Heverlee, Belgium

**Keywords:** ergonomics, fatigue failure theory, musculoskeletal modeling, ergonomic-risk-assessment scale

## Abstract

A new ergonomic-risk-assessment tool was developed that combines musculoskeletal-model-based loading estimates with insights from fatigue failure theory to evaluate full-body musculoskeletal loading during dynamic tasks. Musculoskeletal-modeling output parameters, i.e., joint contact forces and muscle forces, were combined with tissue-specific injury thresholds that account for loading frequency to determine the injury risk for muscles, lower back, and hip cartilage. The potential of this new risk-assessment tool is demonstrated for defining ergonomic interventions in terms of lifting characteristics, back and shoulder exoskeleton assistance, box transferring, stoop lifting, and an overhead wiring task, respectively. The MATE identifies the risk of WMSDs in different anatomical regions during occupational tasks and allows for the evaluation of the impact of interventions that modify specific lifting characteristics, i.e., load weight versus task repetition. Furthermore, and in clear contrast to currently available ergonomic assessment scores, the effects of the exoskeleton assistance level on the risk of WMSDs of full-body musculoskeletal loading (in particular, the muscles, lower back, and hips) can be evaluated and shows small reductions in musculoskeletal loading but not in injury risk. Therefore, the MATE is a risk-assessment tool based on a full-body, musculoskeletal-modeling approach combined with insights from the fatigue failure theory that shows the proof of concept of a shoulder and back exoskeleton. Furthermore, it accounts for subject-specific characteristics (age and BMI), further enhancing individualized ergonomic-risk assessment.

## 1. Introduction

Work-related musculoskeletal disorders (WMSDs), such as lower back and shoulder pain, are frequently observed on the work floor [1]. A combination of excessive workload, non-ergonomic postures, and repetitive movements underlies the development of WMSDs, thereby causing WMSD-related absences from work [2]. To reduce the number of workers affected by WMSDs and the concomitant, absence-related costs, ergonomic guidelines (such as the International Organization for Standardization (ISO) standards 11228 part 1), ergonomic-risk-assessment scales (such as the National Institute for Occupational Safety and Health (NIOSH) [3], and the rapid upper-limb assessment (RULA) [4]) have been developed. For example, the ISO standard 11228 Part 1 advises a reference mass for two-handed lifting under ideal conditions. The NIOSH lifting equation estimates an upper limit for the handled weight during material handling, and the RULA evaluates the risk of WMSDs during an overhead working task. These ergonomic-risk assessments indirectly quantify peak musculoskeletal loading during material handling using the start and/or end posture and estimated joint moments, neglecting the dynamic properties of the motion. This peak in musculoskeletal loading is then combined with task repetition to define the risk of WMSDs. The aforementioned risk-assessment scales may not be the most sensitive to define WMSD risk when compared to other scales, but they are easy to use and still often used in the workplace. Based on these guidelines and risk-assessment scales, ergonomists will then adapt the workplace and working technique or advise using an assistive device to reduce musculoskeletal loading and lower the risk of developing WMSDs. However, many of these guidelines and risk assessments are limited in scalability [5].

When evaluating the risk of developing WMSDs, it is important to account for the combined effect of loading magnitude and task repetition [6]. Indeed, neither the joint kinematics nor joint moments can be directly to musculoskeletal tissue failure, as it is known that the effect of actual peak force and the loading repetition, will determine their combined effect on material strength and hence musculoskeletal failure. Describing this process of fatigue failure [6,7] requires determining for each force magnitude, typically presented as a percentage of the ultimate stress (S), a maximal number of cycles to failure (N), also referred to as the S-N curve (Figure 1). Two new risk assessment scales, the LiFFT [8] and the DUET [9] include task repetition next to musculoskeletal loading when formulating ergonomic recommendations [9]. However, these scales only use indirect and highly simplified estimates of musculoskeletal loading [10,11,12]. Specifically, they evaluate joint moments in an occupational task’s start and/or end position and estimate the joint contact forces [11], thereby highly simplifying complex dynamic lifting tasks and neglecting their dynamic biomechanical properties (i.e., speed and acceleration). As a result, the posture- or moment-based estimates of loading show a discrepancy in joint loading when compared to the joint loading estimated using a detailed musculoskeletal model: Behjati et al., 2019 [13] and Ghezelbash et al., 2020 [14] confirmed that joint contact forces are often underestimated using the more simplified proxy-based approach. In addition, the risk assessments are usually limited to one specific anatomical region (i.e., RULA, DUET evaluating the upper arm) or even one joint (i.e., NIOSH, LiFFT evaluating the L5-S1 joint), thereby neglecting potential concomitant effects of ergonomic interventions in shifting musculoskeletal loading to neighboring joints, potentially overloading them. 

To potentially increase the accuracy of the estimated risk of WMSDs and overcome the limitations of previously developed risk assessments mentioned above, there is a need to quantify full-body musculoskeletal loading in terms of the joint contact forces. In particular, it is relevant to consider musculoskeletal loading at the back (L5) as well as at the elbow, shoulder, hip, and knee, since shoulder and lower back pain are often combined, and a high prevalence of knee and elbow work-related injuries are observed [15]. A full-body musculoskeletal modeling approach combined with 3D motion capture data would allow quantifying full-body kinematics and joint contact forces. Although relatively new in the field of ergonomics, this method is already well established for clinical applications, e.g., to quantify joint contact forces during gait or to evaluate surgical outcomes in clinical populations [16,17]. Indeed, these methods provide the opportunity to accurately estimate full-body musculoskeletal loading in terms of muscle and joint contact forces while accounting for the tasks’ dynamic properties. 

This study aims to develop and demonstrate the added value of a new, ergonomic-risk-assessment tool to evaluate full-body musculoskeletal loading during material handling tasks by combining musculoskeletal-model-based loading estimates with insights from fatigue failure theory. Firstly, the Musculoskeletal-modeling-based, full-body load-Assessment Tool for Ergonomists (the MATE) will be described. The added value of the MATE is then presented based on two case studies: First, the risk of WMSDs during a lifting task is estimated using inertial measurement unit (IMU)-based information on the movement characteristics, and the effect of changing the lifting characteristics on the injury risk is then evaluated. Next, we use the MATE to evaluate the effect of a shoulder and back exoskeleton on the WMSDs risk, demonstrating its added value in exoskeleton assistance tuning. For all case studies, the evaluation of MATE is compared to standard risk-assessment scales.

## 2. Materials and Methods

The developed framework relies on an open-source, musculoskeletal-modeling software, Opensim 4.1 [18], and calculates muscle and joint contact forces using a full-body musculoskeletal model. In short, the musculoskeletal model is first scaled to the worker’s anthropometrics (segment length and mass). After that, segment positions and orientations (derived from IMU data or camera-based 3D-marker coordinates) are used to describe the kinematics of the (material handling) task. Next, the weight of the handled object and the ground reaction forces are used to estimate muscle forces and joint contact forces for all relevant joints; more specifically, L5-S1, hip, knee, shoulder, and elbow. When ground reaction forces cannot be measured, for example, when measuring in the workplace, a ’top-down’ inverse-dynamic approach is used to estimate the muscle and joint contact forces limiting the analyzed joints to the L5-S1, shoulder, and elbow. The muscle forces and contact forces in L5 and the hip are then compared to literature-based, tissue-specific injury thresholds, which account for lifting frequency and task duration.

The estimation of the risk of WMSDs as shown in Figure 2, relies on the standard musculoskeletal-modeling workflow implemented in Opensim 4.1 using a generic, full-body musculoskeletal model (i.e., combining a modified model of the upper extremity [19], the gait 2354 lower extremity [20], and the thoracolumbar spine model of Christophy et al. [21] which model consists of a rigid torso and five lumbar vertebrae of which the kinematics are prescribed as a fraction of the overall lumbar movement). The full-body musculoskeletal model can be scaled to the workers’ anthropometry in two ways: Firstly, the generic model can be scaled based on 3D marker positions measured during a static trial, as described in Delp et al., 2007 [18]. Secondly, a manual scaling option can scale the generic model to the workers’ measured segment lengths. Next, joint angles can be calculated using inverse kinematics based on two inputs: Firstly, based on the 3D coordinates of retroreflective markers positioned on anatomical landmarks measured using a camera-based system, an inverse kinematics approach [22] estimates the full-body joint kinematics by minimizing the weighted difference between measured markers and corresponding markers in the musculoskeletal model. Secondly, based on the orientation data recorded using an IMU system, the newly developed OpenSense tool [23] within OpenSim calculates the kinematics by minimizing the difference between the orientation data from the IMU sensors and the IMU Frames of the calibrated musculoskeletal model. Once the kinematics are defined, an inverse dynamics analysis is used to calculate the full-body joint moments based on the ground reaction forces, the segmental accelerations, positions, and the segments’ inertial characteristics. In the absence of measured ground reaction forces, not uncommon when measuring in the workplace, a ’top-down’ inverse dynamic approach is used to estimate the muscle and joint contact forces only for L5-S1, shoulder and elbow. The weight of the handled object is accounted for by adding half of its weight to the mass of metacarpal III of the right and left hand. Then, the muscle force distribution is calculated that satisfies the external joint moment balance using dynamic optimization while minimizing modeled muscle activation dynamics [24]. Finally, these muscle forces are combined with the joint reaction forces due to ground contact to calculate the joint contact forces at the elbow, shoulder, L5, hip, and knee.

The muscle and joint contact forces are evaluated against tissue-specific injury risk thresholds, experimentally determined for muscles, vertebrae, and hip cartilage. In line with the fatigue failure theory [25], the injury threshold accounts for the force magnitude and loading frequency.

(1) To quantify the risk of WMSDs for individual muscle groups, the MATE evaluates the muscle activations estimated by the musculoskeletal modeling workflow against the maximal acceptable effort (MAE) as defined by Potvin et al., 2012 [26]. The MAE considers the task’s duty cycle (DC), calculated as the fraction of the individual task duration multiplied by its repetition and the total movement time. It is expressed as the percentage of working time spent executing the movement. For example, a worker performs a task of 6 s, five times per minute. This indicates that the worker works for 30 s each minute (=time per cycle), equal to a duty cycle of 50%. This duty cycle can then be used in formula 1 to calculate the MAE.
MAE = 1 − [DC − 1/28800]^0.24,(1)

If half of the modeled activations of agonists (i.e., a functional muscle group that shares a similar primary action) exceeds the MAE threshold, the muscle group is considered at risk as the remaining compensatory reserve force is substantially limited.

(2) To quantify the risk of structural hip-cartilage damage, peak hip stress is evaluated against the endurance limit of hip cartilage, experimentally determined by Riemenschneider et al., 2019 [27] as the loading magnitude that will induce no material failure, independent of the number of loading cycles (Figure 1) [25]. The estimated peak contact force is first divided by the average femoral head area (1610 mm^2^), as described in Teichtahl et al., 2014 [28]. Then, the estimated peak hip stress is evaluated against the endurance limit of hip joint cartilage under repetitive loading, being 5.86 MPa or 9434 N for an average femoral head area of 1610 mm^2^, representative of 30% of the ultimate stress of hip cartilage. All stresses below this limit will not result in injuries independent of the task’s repetition. Therefore, only tasks with hip joint stresses higher than the endurance limit will be classified as being at risk for hip WMSDs. 

(3) To quantify the risk of developing low back pain, the MATE first calculates the failure probability of the vertebrae by evaluating the estimated peak L5 joint contact forces against the ultimate compressive strength of the vertebrae at a specific loading frequency based on the material strength and vertebrae area, as described by Brinckmann et al. [29]. However, the failure probability is not representative of the risk of developing low back pain, and a threshold needs to be defined that accurately discriminates material handling tasks as being either high or low risk of developing low back pain based on the failure probability of the vertebrae. To this end, we used the dataset presented by Zurada et al. [30] that documents peak lumbar flexion position, peak L5-S1 joint moment, task frequency, and 148 material handling tasks that were classified as low- or high-risk based on the associated incidence of low back pain over three years. Using this dataset, we identified the threshold of 10% of the failure probability as this accurately classifies the different material handling (overall classification accuracy = 67%, classification accuracy high-risk tasks = 71%, classification accuracy low-risk tasks = 63%) tasks as high or low risk of developing low back pain (see Section A.1).

## 3. Case Studies

One healthy participant (age: 21 years, weight: 71.4 kg, and height: 175 cm) provided written informed consent before the start of the study, for which the local ethics committee approved all procedures (Ethische commissie onderzoek UZ/KU Leuven, S61611). This participant performed all the tasks associated with the two case studies:

### 3.1. Case Study (1)—Evaluating the Effect of Different Lifting Characteristics on Musculoskeletal Loading and the Risk of Developing WMSDs

Problem statement: The two main determinants in an ergonomic-risk assessment are lifting frequency and the lifted weight. Therefore, these are often adjusted as a preventative strategy to reduce the risk of developing WMSDs. In this case study, we evaluated the effect of different lifting frequencies (3 and 0.2 lifts per minute) and weights (10 kg and 3 kg) during a box-transferring task (lifting a box from the ground, 2 m of walking with the box in the hands, and placing the box on a shelf at shoulder height). For this case study, we simulated the effect of a lower lifting frequency (0.2 lifts per minute) and weight (3 kg) of the handled material separately and the combined effect on the risk of WMSDs in the lower back and muscles.

Methods: Three-dimensional, full-body kinematics were captured with 17 MTx sensors from the MTw Awinda system of Xsens (60 Hz, Xsens, Xsens Technologies, Enschede, NL) and used as input for the musculoskeletal-modeling workflow incorporated in the MATE to calculate the muscle and joint contact forces. Kinematics were kept identical between the four conditions to only evaluate the effect of the exoskeleton assistance and not the consequent effect on task execution. The injury risk associated with the different task executions was then calculated by evaluating the calculated muscle and contact forces against the specified, tissue-specific injury thresholds. Therefore, two lifting frequencies of 3 and 0.2 lifts per minute with a duration of 120 min and two handled weights of 10 kg and 3 kg were assumed. In addition, the NIOSH lifting index [3], the RULA [4], and the REBA [31] risk scores were calculated for all tasks and compared to the MATE’s estimated risk of WMSDs. 

Results All results are shown in Figure 3 and Figure 4.

Reducing the weight of the load from 10 kg to 3 kg decreased all peak joint contact forces. However, the probability of failure and the number of muscle groups at risk (MAE = 25.1%, high repetition and MAE = 61.72%, low repetition; left- and right-sided spine extensor muscles, shoulder flexion muscles, and left shoulder abduction muscles) did not reduce, indicative of the ineffectiveness of affecting the risk of developing WMSDs. This is mainly because the mass of the torso (50.5 kg) predominantly determines the contact and muscle forces, with only minimal effect on the difference in load (7 kg). On the other hand, lowering the task repetition from 3 lifts per minute to 0.2 lifts per minute did reduce the estimated failure probability of the vertebrae by 25%. However, it was still categorized as a high-risk task for developing lower-back pain (exceeding the threshold value of the probability of the failure set at 10%). In addition, lowering the task repetition reduced the risk of WMSDs in three muscle groups (the left- and right-sided shoulder flexion muscles and the left shoulder abduction muscles). 

Discussion: Independent of the different weights and task repetition, this box-transferring task was identified by MATE as a high-risk task for developing lower-back pain. By reducing task repetition, the risk of developing WMSDs was eliminated for the upper limb muscles in particular. However, reducing the weight of the box did not have the same effect. The outcomes from the standard ergonomic-risk-assessment scales, i.e., RULA and REBA, were not affected by reducing the weight of the load or the task repetition (Figure 4). However, the NIOSH lifting index was affected and even decreased through all four ergonomic interventions, therefore indicating a lower injury risk for the lower back after the ergonomic interventions. This is in contrast to the MATE, which indicated a high risk for lower-back pain even after the ergonomic interventions and a lower risk of shoulder injury after reducing task repetition. Furthermore, the MATE could differentiate anatomical regions, indicating that reducing the task repetition did not impact the risk of lower-back pain.

### 3.2. Case Study (2)—Evaluating the Impact of Exoskeleton Assistance on Musculoskeletal Loading and the Risk of Developing WMSDs

Problem statement: As exoskeletons may offer additional protection against musculoskeletal overloading during material-handling tasks, [15,32] there is a rising interest in exoskeletons in the workplace [32]. However, it is unclear how much assistance the exoskeleton should provide during material-handling tasks to affect the injury risk probability. Therefore, we investigated the effect of different back and shoulder exoskeleton assistance levels on the risk of developing WMSDs during a stoop lift and an overhead wiring task, respectively.

Methods: Ten infrared cameras (100 Hz, VICON, Oxford Metrics, Oxford, UK) captured 3D-marker trajectories during the stoop lifting and overhead wiring tasks (pulling laces through holes at a height of 200 cm) while the participant was not wearing any exoskeleton. Ground reaction forces were measured synchronously by two ground-embedded force plates (1000 Hz, AMTI, Watertown, MA, USA). Reflective markers were placed on the participant according to the plug-in gait-marker model [33], extended with a spine-marker model [34]. Three marker clusters replaced the single markers on the limbs. Marker trajectories were used as input for the musculoskeletal-modeling workflow incorporated in the MATE. A lifting frequency of four lifts per minute with a duration of 120 min was assumed for the stoop lifting. A lifting frequency of six handlings per minute with a duration of 120 min was assumed for the overhead wiring task. Then, the effect of a back exoskeleton, providing 30 Nm and 90 Nm of assistance around L5 for the stoop lifting; and the effect of a shoulder exoskeleton, providing 3 Nm and 6 Nm of assistance around the shoulder, were investigated. Under these conditions, kinematics were kept identical to the measured non-exoskeleton kinematics. The assistance provided by the shoulder exoskeleton was applied as a force to the humerus perpendicular to the humerus long axis (more details in van der Have et al., 2022 [35]). The assistance provided by the back exoskeleton was applied as a force to the torso while simultaneously applying half of the assistance force to the left- and right-sided femurs. The mass of the exoskeleton was neglected. These assumptions are inspired by the design of commercially available exoskeletons, such as the Skelex (Skel-Ex, Rotterdam, The Netherlands) for the shoulder exoskeleton and the Laevo (Laevo BV, the Netherlands) for the back exoskeleton [36]. The moment-reducing effect of the exoskeleton was accounted for when solving the muscle force distribution, and the resulting contact forces were then evaluated against the thresholds for the risk of WMSDs implemented in the MATE. In addition, the NIOSH lifting index [3], the RULA [4], and the REBA [31] risk scores were calculated for both tasks.

Results: For the back exoskeleton, peak joint contact forces in the spine and knee reduced more when increasing the assistance (Figure 5). On the other hand, the peak hip-joint contact force was reduced with 30 Nm of assistance but increased with 90 Nm of assistance when compared to the non-exoskeleton condition. Although the failure probability of the vertebrae did decrease by 15% and 35% with 30 Nm and 90 Nm of assistance, respectively, the task was still categorized as a high-risk task for developing lower-back pain (with the probability of failure exceeding the 10% threshold). Although several back-muscle groups were at risk during the non-exoskeleton condition, neither assistance could have a substantial impact (MAE = 34.95%). None of the conditions indicated a risk of developing WMSDs at the hip. The previously developed ergonomic-risk-assessment scales reported in this study, i.e., NIOSH, RULA, and REBA cannot evaluate the effects of an exoskeleton. However, we can compare the risk classification of the previously developed ergonomic-risk-assessment scales with the MATE during the no-exo conditions. The NIOSH defined the stoop as a medium-risk task (LI = 1.3) for WMSDs. It showed a discrepancy in risk classification between the NIOSH and the MATE during stoop lifting. The recommendation made by the RULA score (=4) to change the stoop task soon to reduce the risk of developing shoulder injuries is not in line with the conclusions of the MATE, given that the MATE defines the lower back as at risk and not the shoulder joint.

For the shoulder exoskeleton, peak shoulder-joint contact and muscle forces decreased with increasing assistance (Figure 6). Although forces in several upper limb muscle groups exceeded the injury threshold during the overhead wiring task without the exoskeleton, the exoskeleton assistance did not decrease the number of muscle groups at risk despite the reduced joint contact forces and muscle forces observed (MAE = 22.6%). The failure probability of the vertebrae during the overhead wiring task estimated by the MATE was 0%, and, as a result, there was no increased risk of developing lower-back pain. In addition, no risk of WMSDs for the hip was observed. The previously developed ergonomic-risk-assessment scales reported in this study, i.e., NIOSH, RULA, and REBA, cannot evaluate the effects of an exoskeleton. However, we did compare the risk classification of the previously developed ergonomic-risk-assessment scales with the MATE during the no-exo conditions. The NIOSH defined the overhead wiring task as a high-risk task (LI = 0) for developing lower back injuries, which is not in line with the MATE. The recommendation made by the RULA score (=7) to change the overhead wiring task soon to reduce the risk of developing shoulder injuries is in line with the conclusions of the MATE, as the MATE defined that mainly upper-limb muscles are at risk for developing WMSDs.

## 4. Discussion

The Musculoskeletal-modeling-based, full-body load-Assessment Tool for Ergonomists (MATE) uses a full-body, musculoskeletal-modeling workflow to calculate muscles and joint contact forces and evaluate them against tissue-injury thresholds that account for the loading magnitude and frequency of the specific musculoskeletal tissue. Based on the literature, tissue-specific thresholds were defined for the muscles and the hip cartilage based on experimental material testing. In addition, an injury threshold was determined using a published data set on back pain incidence during material-handling tasks to estimate the risk of developing lower-back pain. The two case studies demonstrated the MATE’s added value in differentiating specific anatomical regions and in evaluating the effects of shoulder and lower-back exoskeleton assistance on the risk of developing WMSDs during occupational tasks.

The MATE allows for the estimation of the impact of altering specific lifting characteristics, i.e., load weight versus task repetition, during a box-transferring task. More specifically, for this task, the reduction of the load’s weight did not significantly reduce the risk of WMSDs, given the dominant effect of trunk weight on the calculated contact forces and injury risk. However, when reducing task repetition, a probability of failure and a reduction in the number of muscle groups at risk were observed. This conclusion is in contrast with the outcome of the NIOSH and the REBA assessments, whose risk scores were mainly affected by a reduction in the weight of the load. More specifically, reducing the weight of the load did reduce the contact forces at L5 by only 700 N and still exceeded the 3400 N NIOSH compression limit. However, following the lifting index of NIOSH, this indicated that reducing the handled weight reduces the injury risk. The use of MATE, therefore, allows for a finer analysis of the factors affecting WMSDs than previous risk-assessment scales as it allows for the discrimination of the effect of handling frequency versus a reduction in handled mass, both relevant and adaptable lifting characteristics in the workplace.

The MATE allows for the identification of the risk of WMSDs for different anatomical regions during occupational tasks. As a result, it indicates that the lower back and shoulder muscles were at risk when performing the box-transferring task. In addition, reducing the lifting frequency for this task alleviated the risk of developing WMSDs for shoulder muscles but did not affect the risk of lower-back pain. Unfortunately, neither of these changes could be identified by the standard risk-assessment scales that cannot differentiate between anatomical regions by definition.

The MATE accounts for the movement execution and dynamic properties of a lifting task, whereas standard ergonomic-risk-assessment scales often use quasi-static models. As a result, musculoskeletal loading during these lifting tasks is often underestimated, resulting in an underestimation of the injury risk. In the current case studies, the MATE defined all three tasks as having a high risk, given that a high risk of WMSDs was observed for the lower back or shoulder muscles. This was not in line with the conclusion of previously developed risk-assessment scales, i.e., the NIOSH, RULA, REBA, and LiFFT. NIOSH and LiFFT categorized the transferring task as low-risk and the REBA and RULA classified it as medium-risk. It is important to note that the MATE defined this task as high-risk for specific anatomical regions, i.e., the shoulder muscles and lower back. Indeed, high contact forces at L5 (4749 N) were observed during the transferring task, which exceeded the NIOSH compression limit of 3400 N, thereby confirming the classification of this task as high-risk, also according to the NIOSH compression limit (3400 N). However, this task was not classified as high-risk when using the NIOSH lifting index. This demonstrates an apparent discrepancy between the NIOSH lifting index (LI = 0.93) and the NIOSH compression limit (4749 N compression > 3400 N). The discrepancy between the NIOSH lifting index and the NIOSH compression limit has been reported before [13,14,37], highlighting the underestimation of injury risk when using already existing ergonomic-risk-assessment scales that use quasi-static models and are calibrated to protect 75% of the female population.

Risk estimation of the MATE on the material handlings described in Zurada et al. has a high overall classification accuracy of 67%, with the classification accuracy of high-risk tasks being 71% and the classification accuracy of low-risk tasks being 63%). Comparing these risk estimation accuracies with the risk estimation accuracies of the NIOSH compression limit of 3400 N on the material handlings described in Zurada et al. showed that using the musculoskeletal-modeling approach in combination with task repetition resulted in more accurate classification of the low- and high-risk tasks based on the reported incidence of lower-back pain, the data for which is available in Zurada (Figure A2 in the Appendix A). Moreover, it is important to consider task repetition when evaluating injury risk as the incidence rate of injury from repetitive motion has not decreased in past years, indicating the need for advanced risk assessments with high prediction accuracies [11]. In addition, the fact that previous ergonomic-risk-assessments have doubtful prediction accuracies was discussed before [38]. Therefore, risk assessments must be improved further to decrease the prevalence of WMSDs in the workplace.

The MATE is one of the first risk-assessment scales allowing us to evaluate the in silico effects of an exoskeleton on the risk of WMSDs on the muscles, lower back, hip, and full-body musculoskeletal loading. Therefore, it can be used to investigate the effect of different assistance levels on full-body musculoskeletal loading. Only recently, the ExoLiFFT [39] was developed as a risk-assessment scale designed to consider the effects of a back exoskeleton on the lower back using joint-moment-based proxies to estimate the exoskeleton’s effect on musculoskeletal loading by subtracting the peak exoskeleton assistance moment from the peak load moment. However, the ExoLiFFT only evaluates the effect of a back exoskeleton on the risk of WMSDs in the lower back. By contrast, the MATE can also be used for shoulder exoskeletons to evaluate the effects on full-body musculoskeletal loading and the risk of developing WMSDs in different anatomical regions. Therefore, the potential effects of exoskeleton assistance on the loading of neighboring joints and the consequent risk of WMSDs can be investigated. In the presented case study, higher levels of assistance from the back exoskeleton reduced musculoskeletal loading in the back more than at lower levels. However, it also increased hip loading without exceeding the risk threshold for hip cartilage. Likewise, the shoulder exoskeleton reduced the musculoskeletal shoulder loading without transferring the load to other joints. Current ergonomic-risk-assessment scales do not allow for such topographic analyses, nor do they allow for an assessment of the effect of different magnitudes of assistance.

The MATE considers individual factors such as age and BMI to determine the ultimate compressive strength of a vertebra [29] and, therefore, the injury risk of the lower back in line with the insights from the fatigue failure theory [25,40]. This is important as an imbalance between loading and load-bearing capacity will impact the risk of WMSDs [41]. However, standard ergonomic-risk-assessment scales do not account for the altered load-bearing capacity based on age and BMI. This has already been identified to underlie incorrect risk categorization [14]. Given that the MATE’s risk threshold for lower-back pain considers both BMI and the worker’s age to determine the fracture limit of the vertebrae, individualized ergonomic advice that is customized for individual factors such as age and BMI becomes feasible. This allows for the definition of individualized injury thresholds even for identical handling tasks, thereby introducing the concept of personalized ergonomic interventions that can further reduce the risk of developing WMSDs. For example, when an old worker performs a similar task as a young worker (and both have the same BMI), the senior worker might be at risk for developing lower-back pain. By contrast, the other, younger worker will not be at risk during the same task, given that their load-bearing capacity is different. Based on their age-specific load-bearing capacities, workers of different ages have different loading capacities resulting in a different allowable number of loading cycles and contact forces in L5 (Figure 7), without considering training or adaptation effects.

Although there is an added value of the newly developed risk-assessment scale when compared to previously existing ergonomic-risk-assessment scales, it is important to outline the following limitations of the presented study. First, the database used to determine the threshold for developing lower-back pain existed of 148 material-handling tasks with mainly an upright, standing posture. As a result, the probability-of-failure threshold used to classify high and low risk using the MATE—at least in theory—can only be considered valid for tasks with an upright, standing posture. Therefore, further calibration and validation, including various occupational tasks, is required. However, to our knowledge, the publicly available dataset can relate musculoskeletal loading to the risk of developing lower-back pain. Second, although full-body joint contact forces can be estimated using the MATE, the risk of WMSD estimation is currently limited to the hip and back joints and the muscle structures. This is because the current literature only describes the interaction effect of loading magnitude and frequency on material strength for these biomaterials. Therefore, further experimental material testing is warranted to document the fatigue failure theory further. Thirdly, the current implementation in MATE does not allow for the evaluation of the impact of cumulative loading on the risk of WMSDs, nor the effects of rest or healing. Both factors are known to influence the risk of WMSDs [40]. Fourth, movement execution was kept identical between the conditions within the case studies in order to focus on the effect of the ergonomic intervention. However, movement execution could change when ergonomic interventions are applied in the workplace, limiting the representativeness of the case studies. Fifth, the estimated absolute contact forces were evaluated against maximal fatigue failure thresholds to determine the injury risk, independent of validation of the estimated contact forces. However, based on contact forces, the risk threshold of developing lower-back pain was calibrated using the dataset from Zurada, and we were able to establish that the risk-threshold accuracy is high. This way, the calibration procedure used to determine the risk threshold canceled out potential under- or over-estimated absolute values. This highlights the need to validate the applied calibration procedure further using an independent dataset.

## 5. Conclusions

In conclusion, we propose using the MATE as a risk-assessment tool based on a full-body, musculoskeletal-modeling approach combined with insights from the fatigue failure theory which could potentially improve future ergonomic advice and further impact WMSDs. MATE allows for region-specific risk assessments while accounting for the effect of assistive-device assistance in the case of back and shoulder exoskeletons. Furthermore, it accounts for subject-specific characteristics (age and BMI), further enhancing the individualized ergonomic-risk assessment.

## Figures and Tables

**Figure 1 ijerph-20-01507-f001:**
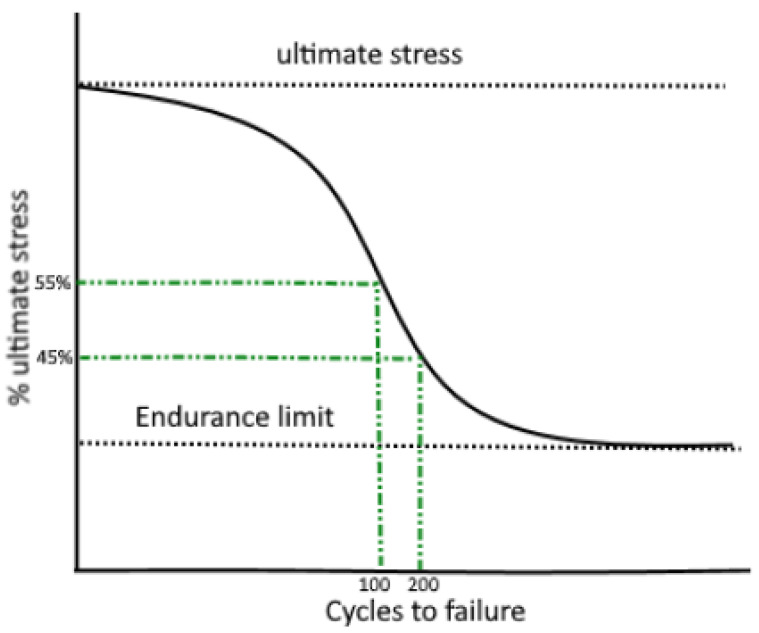
S-N curve for a biomaterial representing the basis of the fatigue failure theory. S is the percentage of ultimate stress and N is the number of cycles to failure.

**Figure 2 ijerph-20-01507-f002:**
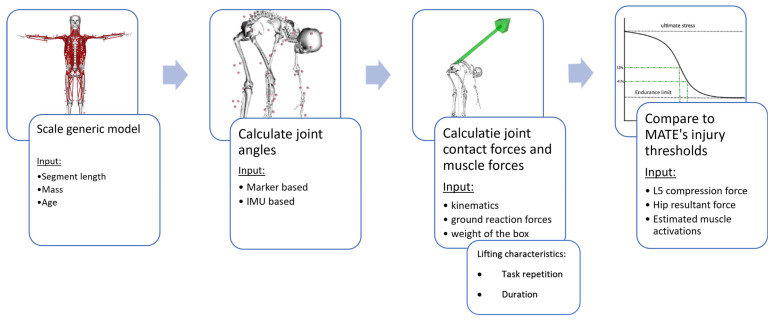
General overview of the MATE workflow.

**Figure 3 ijerph-20-01507-f003:**
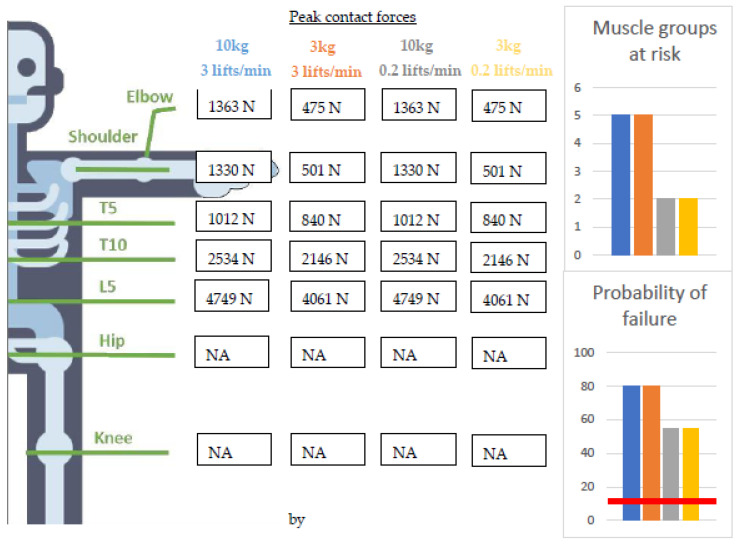
Peak joint contact forces in the different joints calculated with the musculoskeletal-modeling workflow and compared to the different tissue-specific injury thresholds implemented in the MATE for the different working conditions. Values above the red line are high risk. Ground reaction forces were not considered, and a top-down approach was used. As a result, the hip and knee joints could not be analyzed (NA).

**Figure 4 ijerph-20-01507-f004:**
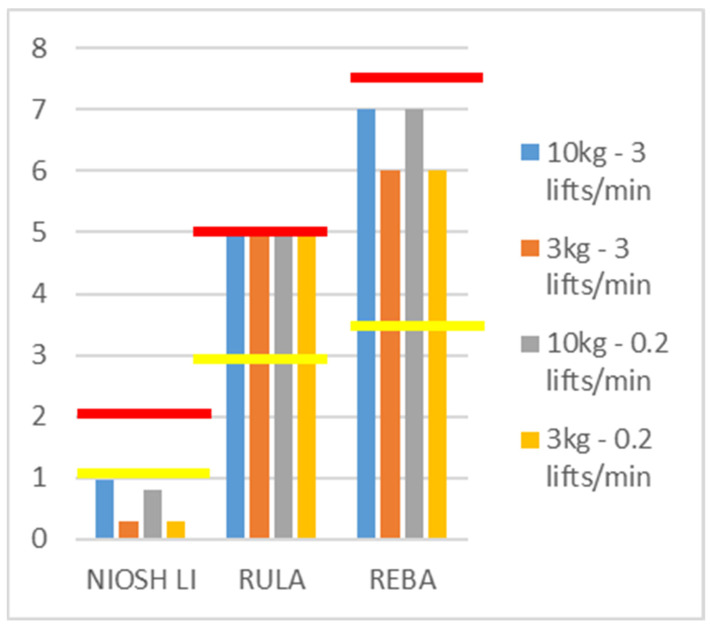
NIOSH Lifting Index, RULA, and REBA scores estimated for the different working conditions. Values below the yellow line are low risk, values between the yellow and red line are medium risk, and values above red line are high risk, according to the specific guidelines.

**Figure 5 ijerph-20-01507-f005:**
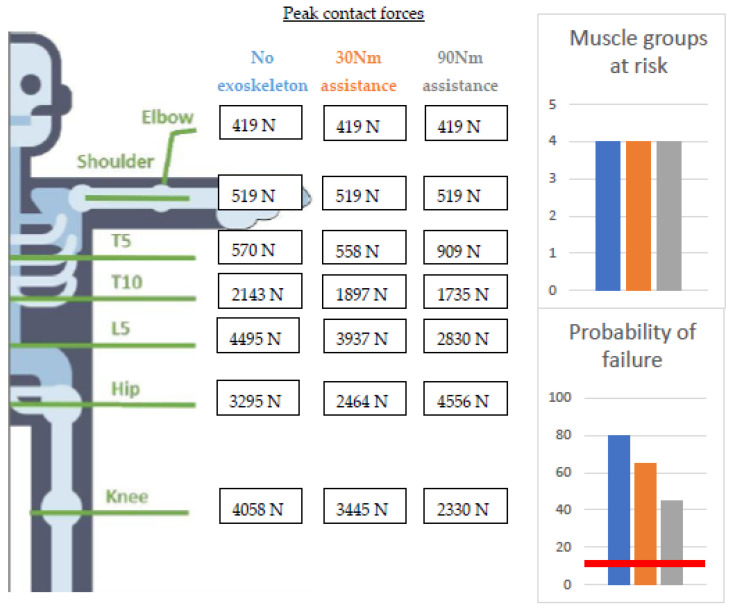
Peak joint contact forces in the different joints calculated with the musculoskeletal-modeling workflow and compared with the different tissue-specific injury thresholds implemented in the MATE for the back exoskeletons. Values above the red line are high risk.

**Figure 6 ijerph-20-01507-f006:**
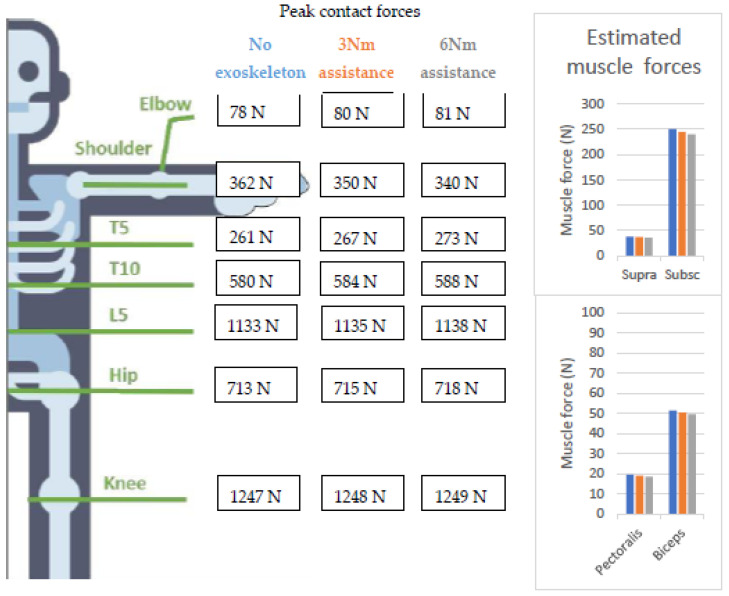
Peak joint contact forces in the different joints calculated with the musculoskeletal-modeling workflow and compared with the different tissue-specific injury thresholds implemented in the MATE for the shoulder exoskeletons. The probability of failure is not shown as it was equal to 0%, indicating this task has a low risk of developing low back pain. In addition, muscle groups at risk are not shown as the exoskeleton did not affect this number. However, estimated muscle forces of the m. Supraspinatus (Supra), m. Subscapularis (Subsc), m. Pectoralis major (Pectoralis), and the m. Biceps brachii (Biceps) did decrease with increasing the exoskeleton assistance.

**Figure 7 ijerph-20-01507-f007:**
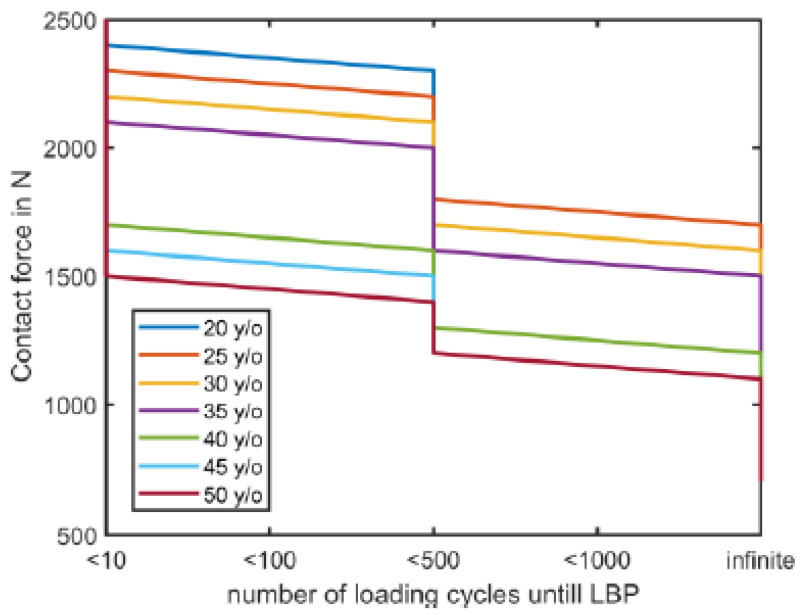
Maximal number of loading cycles prior to lower-back pain (LBP) based on the L5 contact force for different workers of different ages.

## Data Availability

Not applicable.

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
