# Peer review of "Musculoskeletal-Modeling-Based, Full-Body Load-Assessment Tool for Ergonomists (MATE): Method Development and Proof of Concept Case Studies"

_ijerph, 2023, doi:10.3390/ijerph20021507_

Round 1

Reviewer 1 Report

The purpose of this research is to improve upon ergonomic assessment methods by incorporating metrics computed from a musculoskeletal model developed in OpenSim. The method claims to overcome limitations with a few common assessment methods including NIOSH RLE, RULA, REBA. In principle the approach that is described in this work could provide certain advantages over existing, less sophisticated methods, however there are limitations that haven’t been adequately addressed with this approach. This work is not very clear about how this method would provide better recommendations for improving ergonomics in the workplace. Additionally, the criticism of other methods used as comparison are not completely founded on good science. I suggest that the authors review the substantial body of work that references methods like the NIOSH RLE to become more familiar with the strengths and limitations. This would allow for a greater understanding of how the MATE can enhance or replace these methods. One major, limitation and concern about the authors’ presentation of MATE is that it’s purely a biomechanical model, meaning it doesn’t account for other factors that risk assessment methods like NIOSH RLE have. Given the amount of literature, specifically about low back pain and non-work related activities, the fatigue failure theory that is used as the foundation of the LiFFT and DUET, and even the MAE remains limited. Furthermore, validation of these methods is largely lacking in the literature. Furthermore, the data presented in this manuscript create questions about how the experiments were conducted and analyzed. The pdf. of the manuscript contains many comments and questions about the work and I summarize my key observations and concerns here.

1.      In principle, more information about the musculoskeletal stress induced on the body during occupational activities should inform more sensitive risk assessments at an individual level. I think this is meritorious and a good goal to work towards. However, ignoring the challenges with implementing a workflow for risk assessment as described by MATE is short sighted and should be at least acknowledged by the authors. Furthermore, there are limitations with the MATE that are not discussed, which may lead a reader to believe that there aren’t any limitations.

2.      In order to perform a more critical evaluation of the value added by an OpenSim model for ergonomic assessments, a more comprehensive sensitivity analysis of how results derived from OpenSim and then converted into risk is necessary. The case studies that are reviewed in the work are not sufficient and there is insufficient detail about the experimental condition to completely assess the validity of results that are presented. As one example, the joint compression values for the elbow and shoulder for all of the trials shown in Figure 5 is the same. This suggests that perhaps only 1 trial was performed and the model was manipulated to produce different outputs for the joints that would have been most likely affected by the task conditions.

3.      The task variables chosen for the study (load weight and frequency) were such that they didn’t exploit the range of risk. This may lead the readers to believe that these methods are inaccurate. It is important to state why the task variables were chosen and what the expectations are based on your knowledge of the strengths and limitations of the methods that are being used. 

Overall, I believe that the MATE can offer new insights into MSD development and prevention that existing methods for ergonomics risk assessment can not and the authors should consider revising the manuscript so readers become more aware of strengths and limitations of the MATE and the methods chosen for comparison. Further, the feasibility of mocap and GRF data collection outside a controlled laboratory are not even mentioned here, but could be a significant impediment for widespread adoption and use by practitioners. How can the work that is presented here contribute to a paradigm shift towards more sophisticated methods? 

Author Response

Dear reviewer

Please see the attachment for the point-by-point responses. 

Thank you for reviewing

Reviewer 2 Report

Mihnea-Ion MARIN

E-mail: mihnea_marin@yahoo.com,   mihnea.marin@edu.ucv.ro

 Thank you for the opportunity to review this article.

Peer-review report 2086724

The aim of the study Musculoskeletal modeling-based full body load Assessment Tool for Ergonomists (MATE): Method development and proof of concept case studies, written by authors: Arthur van der Have, Sam Van Rossom and Ilse Jonkers, was to present a new ergonomic risk assessment tool to evaluate full-body musculoskeletal loading during material handling tasks, by combining musculoskeletal model with failure theory.

The ergonomic tool combine in a complicated way, 3D motion capture date with  the fatigue failure theory.

1. In the introduction, the authors make numerous references both to scientific articles in which kinematic parameters of repetitive work activities have been analysed and to scales specifying risk levels in ergonomics and make recommendations for the application of specific regulations.

2. The Materials and Methods chapter is devoted to detailing the methodology on which the ergonomic tool was built and how hazardous muscle activities were quantified, the risk of structural hip cartilage damage and the risk of developing low back pain.

3. The results were replaced by two case studies

4. Even though the discussions are detailed and extensive, I think the article is written in a way that is difficult for the reader to follow, with many details that make reading difficult.

5. In conclusion the authors propose using the MATE as a risk assessment tool based on a full body musculoskeletal modeling approach combined with insights from the fatigue failure theory.

A few observations related to form:

Line 52: ”nor” should be replaced by ”not”;

Line 55: revise spacing;

Line 89: This study aims is to develop...;

Line 160: ”[…]which is 5.86 MPa or 9434N,[…]” MPA is the unit of pressure and N is the unit of force. I don't understand why the authors consider 5.86 Mpa to be equal (equivalent) to 9434 N.

Line 244: “maker” should be replaced with “marker”;

Line 580: the year 2016 should be bold.

In my opinion, the MATE tool does not bring any novelty compared to the BoB software (Biomechanics of Bodies- https://www.bob-biomechanics.com/bob-4-ergo/). The paper shows a keen interest of the authors to develop an interesting and complex tool and can be presented as a contribution of the authors in the field of ergonomic evaluations.

Author Response

(The authors gave the same response as above.)

Round 2

Reviewer 2 Report

Thanks to the editors for the opportunity to read the new form of the manuscript.

Thanks to the authors for the coverletter.

I believe that this revised form of the manuscript can be published in IJERPH, Injury Prevention and Rehabilitation section.